# Characterization of Botanical Origin of Italian Honey by Carbohydrate Composition and Volatile Organic Compounds (VOCs)

**DOI:** 10.3390/foods11162441

**Published:** 2022-08-13

**Authors:** Raffaello Tedesco, Elisa Scalabrin, Valeria Malagnini, Lidija Strojnik, Nives Ogrinc, Gabriele Capodaglio

**Affiliations:** 1Department of Environmental Sciences, Informatics and Statistics, University of Venice, Ca’ Foscari, Via Torino 155, 30172 Venice Mestre, Italy; 2Centro Ricerca e Innovazione, Fondazione Edmund Mach (FEM), Via E.Mach 1, San Michele all’Adige, 38010 Trento, Italy; 3National Research Council, Polar Science Institute, Via Torino 155, 30172 Venice Mestre, Italy; 4Department of Environmental Sciences, Jožef Stefan Institute, 1000 Ljubljana, Slovenia

**Keywords:** Italian honey, melissopalynological analysis, carbohydrates, apple–dandelion, PLS-DA

## Abstract

Honey is a natural sweetener constituted by numerous macro- and micronutrients. Carbohydrates are the most representative, with glucose and fructose being the most abundant. Minor honey components like volatile organic compounds (VOCs), minerals, vitamins, amino acids are able to confer honey-specific properties and are useful to characterize and differentiate between honey varieties according to the botanical origin. The present work describes the chemical characterization of honeys of different botanical origin (multifloral, acacia, apple–dandelion, rhododendron, honeydew, and chestnut) produced and collected by beekeepers in the Trentino Alto-Adige region (Italy). Melissopalynological analysis was conducted to verify the botanical origin of samples and determine the frequency of different pollen families. The carbohydrate composition (fourteen sugars) and the profile of VOCs were evaluated permitting to investigate the relationship between pollen composition and the chemical profile of honey. Statistical analysis, particularly partial least squares discriminant analysis (PLS-DA), demonstrates the importance of classifying honey botanical origin on the basis of effective pollen composition, which directly influences honey’s biochemistry, in order to correctly define properties and value of honeys.

## 1. Introduction

Honey is a natural complex product and the oldest sweetening agent widely consumed around the world both for attractive taste and nutritional value and even for its health benefits [1,2].

According to Codex Alimentarius published by the Food and Agriculture Organization (FAO) and Directive 2001/110/EC, honey is legally defined as “the natural sweet substance produced by *Apis mellifera* bees from the nectar of plants (blossom honey) or from secretions of living parts of plants or excretions of plant-sucking insects on the living parts of plants (honeydew honey), which the bees collect, transform by combining with specific substances of their own, deposit, dehydrate, store and leave in honeycombs to ripen and mature” [3]. The different honey types may be labelled with the floral origin if the product comes wholly or mainly from the indicated source and reflects its characteristics [4]. European regulation, however, does not define the characteristics of honey floral origin, causing uncertainties in the trading of these products. Some countries have established national regulations or technical criteria to fill this gap (Table 1) [4]. Conventionally, determination of the botanical origin of honey is conducted by pollen analysis through the melissopalynological method which consists in the determination of the share of pollen grains in a honey sample, gaining evidence of the botanical genus of the plants that the honey bees visited [5]. Due to the lack of regulation regarding monofloral honey and the different geographical areas in which they are produced, honeys could present a very variable pollen composition and different relative abundances, even of the characteristic plant genus. Many monofloral honeys characterized by underrepresented pollens (e.g., acacia or dandelion honeys) could, indeed, present high levels of other pollens, which can vary a lot depending on the geographical origin and season [6]. *Castanea* pollens, for instance, are very common in both monofloral and multifloral Italian honeys and could be present even in high percentages due to the wide distribution of this plant genus [7]. Chestnut honey, moreover, must be characterized by at least 90% of *Castanea* pollens, therefore, honeys with lower levels are classified as multifloral or as other monofloral honeys depending on their composition [4,8,9]. Multifloral honeys, thereafter, present a highly variable percentage of pollen grains and, generally, are the remaining honeys which could not be classified as monofloral [10]. Since the botanical origin influences the biochemistry of honeys, this kind of classification determines the heterogeneity in the composition of all honey typologies. Therefore, to correctly define properties and value of honeys, the botanical origin should be related to the chemical characterization.

Honey regulations, including Directive 2001/110/EC, include the definition of quality compositional criteria, such as sugar content, moisture content, acidity, electrical conductivity, diastase activity, and hydroxymethylfurfural (HMF) content [12]. Sugars (mainly fructose and glucose) and water are the most abundant components of honey [13,14,15], which, however, consists of about 180 different constituents. The minor components of honey like minerals, vitamins, amino acids, enzymes, and volatile organic compounds can be used as fingerprints to differentiate between honeys by botanical and geographical origin and describe their quality [16]. These compounds are affected by such factors as botanical and geographical origin, beekeeping activity, seasonal and environmental conditions [4,17,18]. In order to highlight the relationship between pollen composition and biochemistry of honey and investigate the effectiveness of honey classification based on European regulations [4,8,9,11,12], forty-eight Italian honey samples of different botanical origin were analyzed.

Multifloral and monofloral (acacia, apple–dandelion, chestnut, rhododendron) samples produced in northeastern Italy (Trentino Alto-Adige region) were considered in order to trace their melissopalynological and chemical profile. To our knowledge, this is the first study in which apple–dandelion honey, which is typical of northern Italian mountain regions, is characterized. The analysis of sugars, particularly glucose, fructose, sucrose, and di- and trisaccharides, and of VOCs (mainly carboxylic acids, aldehydes, alcohols, and terpenes) was performed. In this study, previously published data about sugars [19] for 33 of the 47 samples here considered and new melissopalynological, sugar, and VOC results are discussed together. These parameters have been shown to play an important role in the identification of floral [20,21,22] and geographical origin [23,24,25], significantly contributing to the classification of honeys. Melissopalynological analysis permitted to verify the botanical origin and identify all the pollens present in the samples in order to trace a correlation with chemical parameters. The data were processed by chemometric methods, particularly by partial least squares discriminant analysis (PLS-DA). This methodology allowed highlighting the importance of honey classification based on pollen content in both monofloral and multifloral honeys. Pollen composition, indeed, directly influences the biochemistry and the organoleptic properties of honeys and should be considered when evaluating their effective commercial value.

## 2. Materials and Methods

### 2.1. Honey Collection and Sampling

Forty-eight samples of *Apis mellifera* honeys were used in the present study. They were obtained directly from beekeepers and collected during the 2017 and 2018 from different locations across the Trentino Alto-Adige region, northeastern Italy. Collection sites were chosen to include different botanical origin within the selected region. The sample set included 24 multifloral and 23 monofloral honeys. The monofloral honey samples were three acacia, three apple–dandelion, seven rhododendron, three honeydew, and seven chestnut ones. The number of samples of each floral group was representative of the kind of honeys produced in the studied area and of the specific conditions of the year of samples collection. Some samples, as acacia and apple–dandelion honeys, were influenced by the frequent bad weather in 2018; short flowering of acacia and dandelion species lead often to the confluence of these honeys to multifloral [26].

In Table 2, the descriptive characteristics of honey samples such as sample code, botanical and geographical origin, and harvest year are summarized.

Collection of samples was carried out making sure that they were representative of the honey lot. Particularly, samples were composed by honey collected from three different beehives of the same production lot. Honeys were extracted by centrifugation, collected in glass jars, and immediately stored at +4 °C in a dark place in order to preserve the original characteristics of the samples, and then were held until required for analysis. Treatment and preparation of the samples were carried out following the recommendations of the Harmonized Methods of the International Honey Commission [27]. Before analysis, all the liquid honey samples were softly homogenized, while the crystallized honeys were previously softened by heating in a thermostatic bath at a temperature ranging from 35 °C to 38 °C.

### 2.2. Chemicals and Reagents

All amino acid standards had a high purity (>98%). D-glucose and D-fructose were supplied by Sigma-Aldrich (Saint Louis, MO, USA). D-sucrose was obtained from Fluka (Ronkonkoma, NY, USA). D-turanose, D-melibiose monohydrate, palatinose hydrate, kojibiose, nigerose, erlose, lactose, lactulose, isomaltotriose, D-raffinose pentahydrate, and D-melezitose were purchased from Santa Cruz Biotechnology, Inc. (Heidelberg, Germany). Sulfuric acid, potassium hydroxide, sodium hydroxide, and sodium chloride were obtained from Sigma-Aldrich (Buchs, Switzerland). Kaiser’s glycerol gelatin for microscopy was obtained from Merck (Merck KGaA, Darmstadt, Germania). Fuchsin Basic alcoholic solution 0.1% and diethyl ether were purchased from PanReac AppliChem (Panreac Química SLU, Barcellona). Ultrapure water (18.2 MΩ/cm, 0.01 TOC) was produced by a Purelab Ultra System (Elga, High Wycombe, UK).

### 2.3. Melissopalynological Analysis

The botanical origin of the honey samples was assigned using melissopalynological analysis following the method of Louveaux et al. [6].

Briefly, the honeys were accurately weighted (10 g) directly into centrifuge tubes with conical ends and diluted with 20 mL of warm ultrapure water (35 °C). The samples were centrifuged for 15 min at 3000 rpm at room temperature (Centrifuge 5430R, Eppendorf AG, Hamburg, Germany). The supernatant was discarded and the deposit was rinsed and centrifuged again with 10 mL of ultrapure water to remove sugars. The washed deposit was layered on a slide, dried at 35 °C, and finally covered with a solution of glycerin–gelatin.

The pollen grains were classified and counted on a square of 18 × 18 mm using a microscope with a magnification of 400×, 600×, and 1000×. According to melissopalynology [6], the floral origin of honey is classified by the percentage of pollen frequencies. Pollens in honey are differentiated according to their contribution to the total content: the predominant pollen (>45% of the pollen grains counted), the accompanying or secondary pollen (16–45%), important minor pollen elements (3–15%), and minor pollen (<3%). According to European national legislations [4] and melissopalynological methods [6], the requested percentage of a specific pollen to classify monofloral nectar honeys is different for each pollen. *Robinia* pollens are underrepresented and can vary between 15% and 30%; on the contrary, Fagaceae are normally overrepresented, therefore, only honeys containing at least 90% of these pollens should be classified as chestnut honey. Honeys containing from 30% to 60% of Ericaceae pollens could be considered monofloral rhododendron honeys; the dandelion honey is classified as monofloral honey with a low percentage of Asteraceae pollens (5–15%). Italian dandelion honeys are generally contaminated with Salicaeae [9]. In apple–dandelion honey, Rosaceae pollens are underrepresented, with percentages lower than 30%. Honeydew honeys are identified using the ratio between honeydew elements (HDE) and the total frequency of pollen from nectar (P) that must be higher than 3 in honeydew honeys [6].

### 2.4. Sugars Analysis

The carbohydrates determination was carried out according to an already published method [19]. Of the 47 samples here globally considered, sugars were determined on 14 samples, while on the others, carbohydrate data had already been published [19]. Briefly, 50 mg of a honey sample were weighted in a 50 mL volumetric flask, spiked with the internal standard (^13^C_6_-glucose at 1 mg/L^−1^), and diluted with ultrapure water until a final concentration of 0.1 mg/mL^−1^. Carbohydrates analysis was carried out using an ion chromatograph (Thermo Scientific™ Dionex™ ICS-5000, Waltham, MA, USA) coupled to a single quadrupole mass spectrometer (MSQ Plus™, Thermo Scientific™, Bremen, Germany) and using a CarboPac PA10™ as a column (Thermo Scientific, Waltham, MA, USA, 2 mm × 250 mm, 10 µm) and a CarboPac PA10™ as a guard column (2 × 50 mm) to perform chromatographic separation of each sugar. Carbohydrate data are expressed in g/100 g.

### 2.5. Volatile Organic Compounds Analysis

#### 2.5.1. Extraction Conditions and Sample Preparation

Headspace solid-phase microextraction (HS-SPME) was achieved using a divinylbenzene/carboxen/polydimethylsiloxane (DVB/CAR/PDMS) fiber, 50/30 µm-thick, 1 cm, Stableflex/SS, 23Ga (Supelco, Bellefonte, PA, USA). The fibers were preconditioned for 30 min in the gas chromatograph’s (GC) injector at the temperature of 270 °C as recommended by the manufacturer. The honey samples were processed following the previously reported procedures [28,29] with some variations in order to optimize the method. Particularly, the equilibration and extraction steps were fine-tuned in terms of time and temperature, to maximize compounds’ recovery. The analytical procedure was optimized using a multifloral honey.

The samples were weighted (1 g) in a 10 mL clear glass vial (with silicone/PTFE septa) and combined with 1 g of ultrapure water and 30% (*w*/*w*) of sodium chloride (NaCl). NaCl addition is used to favor the volatilization of organic compounds due to the decrease in the partition coefficient between the liquid and gas phases, allowing more analytes to readily partition into the headspace. The vials were then hermetically sealed. The samples were automatically mixed and heated at 60 °C for 15 min to achieve the equilibration phase. The fibers were directly introduced into the headspace of the vial for the extraction by exposition for 90 min at 60 °C; the efficiency of the extraction process was improved by continuous stirring. The extracted compounds were thermally desorbed in the GC injector port in splitless mode at 250 °C for 1 min. The analytical procedure was optimized using a multifloral honey.

#### 2.5.2. Gas Chromatography–Mass Spectrometry

The GC–MS analysis was carried out using an Agilent 7890B gas chromatograph (Agilent Technologies, Palo Alto, CA, USA) coupled to a single quadrupole mass spectrometer Agilent 5977A inert MSD (Agilent Technologies, Palo Alto, CA, USA) and equipped with an autosampler (MPS, Gerstel, Mülheim, Germany). The instrumental method was based on the protocol of Robotti et al. (2017) [28]; the chromatographic ramp was optimized in order to achieve the best separation possible and enhance the intensity of compounds. The GC analysis was conducted using a VF-WAXms column (30.0 m length × 0.25 mm I.D. × 0.25 µm film thickness, Agilent Technologies, Palo Alto, CA, USA). The temperature program was as follows: the initial oven temperature was set at 40 °C, then increased at a velocity of 3 °C/min^−1^ until 140 °C and held at this temperature for 10 min; the oven was heated at 5 °C/min^−1^ until 230 °C and held at this temperature for 2 min; then, the temperature was increased at 5 °C/min^−1^ until 250 °C. The carrier gas (helium) flow rate was 1.0 mL/min^−1^. Mass spectrometry analysis was performed in the full-scan acquisition mode and the mass range considered was from 35 to 300 *m*/*z*. The electron impact ionization system was employed at 70 eV at a temperature of 240 °C. The chromatographic analysis lasted 67 min. Each sample was analyzed in three replicates. The identification procedure was performed by comparing retention times and mass spectra with the NIST 14 Mass Spectral Library (Agilent Technologies, Santa Clara, CA, USA). Eighty VOCs were identified, and their retention time and molecular weights are listed in Appendix A. Of the 80 identified compounds, 24 were selected, a larger part of the samples.

Instrumental repeatability was evaluated by the analysis of seven replicates using a multifloral honey which contained all the compounds of interest and evaluating the relative standard deviation (RSD%) of the peak areas. The mean area values, standard deviations, and relative standard deviations of these 24 compounds are reported in Appendix A; the RSD% was always lower than 11%, except for octane and tetradecanoic acid, which showed values of 18% and 16%, respectively. The area values of all the samples used for the following statistical elaboration are the mean values of three measurements.

### 2.6. Statistical Analysis

Statistical analysis was used to highlight the relationship between chemical composition and the floral origin of honeys in order to identify their specific characteristics.

Metaboanalyst 5.0 software ((Mcgill University, Montreal, QC, Canada) was employed for statistical elaborations [30,31]. Data under the limit of detection were replaced with randomized values between zero and half of the detection limit for each feature calculated as three times the tenth part of the smallest detectable area. Data were normalized, and the analysis of variance (one-way ANOVA test) and Student’s *t*-test were carried out; differences were considered significant at a probability level of *p* < 0.05. PLS-DA was performed as the classification method to model the relationship between pollens and chemical analysis.

## 3. Results

### 3.1. Melissopalynological Data

Honeydew honey, which is produced when bees harvest excretions of plant-sucking insects on the living parts of plants, is identified by the HDE/P ratio. The HDE/*p*-values of honeydew honeys ranged between 3.13 and 7.61 and are reported in Table 3.

Monofloral samples were classified into five groups based on their botanical origin: acacia, apple–dandelion, rhododendron, honeydew, and chestnut honey. Dandelion (*Taraxacum officinale*, Asteraceae) is a botanical species widely spread in many countries of Central Europe, and it is also present in the Alpine area of Italy. However, as reported above, because the Asteraceae pollen is strongly underrepresented, important amounts of honeys of different botanical origin contribute regularly to the dandelion honey composition [32]. In view of plant composition and contemporary flowering of dandelion and apple tree in the Trentino Alto-Adige region between the Val di Non and Val d’Adige areas, the dandelion nectar is normally associated with apple pollen; therefore, apple–dandelion is a typical honey produced in this area. In acacia honey, the main range of *Robinia* (Fabaceae) pollens was from 17% to 19%. Apple–dandelion honeys contained between 12% and 22% of *Malus*/*Pyrus* pollens (Rosaceae) and from 2% to 23% of Asteraceae; the samples highly differed for the content of other minor pollens, varying between high percentages of Fagaceae and Salicaceae, Fabaceae, Vitaceae, and Hippocastanaceae. The Ericaceae pollens in rhododendron honey ranged from 31% to 91%; two samples presented relevant percentages of Fagaceae (23.0–64.2%). The Fagaceae pollens content in chestnut honey ranged between 90% to 98%. The minimum and maximum percentages of pollens in monofloral honeys according to two classifications (family and genus) and their statistically significant differences in the various floral types are reported in Table 4. Multifloral honeys showed a wide variability in pollen composition, which reflects the main plants and floral species present in the investigated area and contributing to the honey production. The melissopalynological composition presented a generally predominant presence of Fagaceae and Ericaceae and minor values of Rosaceae and Fabaceae; however, many samples also showed the presence of pollens belonging to other different classes. The complete results of melissopalynological analysis are reported in Appendix A.

### 3.2. Sugar Content

Fourteen carbohydrates, including two monosaccharides (glucose and fructose), six disaccharides (sucrose, melibiose, kojibiose, turanose, palatinose, nigerose, lactose, and lactulose), and four trisaccharides (melezitose, raffinose, isomaltotriose, and erlose), were determined in the honey samples. The carbohydrate concentrations of all the 47 samples are reported in Table 5; 34 of these had already been reported [19].

Particularly, fructose is the principal monosaccharide present in all the floral honey samples whilst glucose is the second most important simple sugar. According to the compositional criteria of honey, the sum of fructose and glucose must be higher than 60% while in honeydew honey, the percentage must exceed 45% [4,12]; in our samples, the sum of fructose and glucose was higher than 60% and 50% in blossom and honeydew honeys, respectively. Eleven samples (M37, M38, M39, M41, M42, M5-18, M6-18, M19-18, C40, C8-18, C12-18) had slightly lower fructose and glucose percentages than 60%, mainly due to a low glucose content.

As reported in Directive 2001/110 EU [12], the major part of monofloral genuine honey should not exceed 5% of sucrose content. The distribution of sucrose content in all the samples investigated was very broad, and its level in honey depends on the activity of specific enzymes (α- and β-glucosidase, α- and β-amylase, and β-fructosidase) which hydrolyze it into glucose and fructose; monosaccharides are successively used to synthesize new carbohydrates [20,33]. Our sucrose results are in accordance with those obtained in surveys conducted on honey samples produced around the world with the same botanical origin; the highest value, 2.5%, was detected in rhododendron honey [21,34,35,36,37,38,39,40,41,42,43,44,45,46,47,48,49].

Melibiose, kojibiose, turanose, palatinose, lactose, lactulose, and nigerose were present in almost all the multifloral and chestnut honeys considered here. These carbohydrates are essentially formed by units of glucose and fructose differentiated by a glucosidic bond in various positions and spatial configurations [33]. The two disaccharides which showed the highest mean abundances were lactose and turanose. Chestnut honeys exhibited the highest mean value of lactose (2.14 g/100 g) in comparison with multifloral honeys (1.71 g/100 g) and previously reported data (Table 5). Literature data report a very low concentration of lactose in honey, approximately 0.01%, and it was hypothesized that it might be useful in honey characterization along with other sugars [40]. The mean turanose value was comparable in multifloral and chestnut honeys in comparison with the previously reported data [19]. However, chestnut honey exhibited the highest mean concentration (1.71 g/100 g). Among the trisaccharides, erlose was the most abundant, with a higher mean value in multifloral honeys than in chestnut honeys. This sugar originates from sucrose through the metabolism of honeybees and its concentration is modified during storage by enzymatic activity (α-glucosidase) [41]. The statistically significant differences observed in sugar content in the various floral types are evaluated in the Discussion section (Section 4) together with other analytes.

### 3.3. Volatile Organic Compounds

Eighty VOCs were identified in the samples, including carboxylic acids, aldehydes, alcohols. However, twenty-four volatile organic compounds (octane, γ-terpinene, octanal, 2-nonanone, nonanal, acetic acid, furfural, decanal, benzaldehyde, lilac aldehyde C, linalool, hotrienol, 3,7-dimethyl- (hotrienol), benzeneacetaldehyde (phenylacetaldehyde), terpineol, heptanoic acid, phenylethyl alcohol, octanoic acid, nonanoic acid, thymol, n-decanoic acid, geranic acid, benzoic acid, dodecanoic acid, and tetradecanoic acid) were detected in almost all the samples. The remaining volatile substances were found only in a few samples. The sixteen most intense (a) and less intense (b) VOCs found in honey samples are reported in Figure 1; the remaining eight compounds, which showed low intensity and high standard deviation in all the floral honey groups, are reported in the Appendix A. The most intense compound was benzaldehyde in all the monofloral and multifloral samples, with a higher mean intensity in apple–dandelion honeys (Figure 1); hotrienol showed the second highest abundancy, especially in the acacia samples, while the lowest value was registered for chestnut honey. Benzoic acid showed high intensity in all the monofloral and multifloral samples; the other relevant compounds were furfural, nonanoic acid, and octanoic acid (Figure 1).

The most significant compounds are evaluated in detail in the Discussion section (Section 4), separately for each floral group.

## 4. Discussion

### 4.1. Acacia Honeys

Pollens of Fabaceae, the family to which *Robinia pseudoacacia* belongs, were statistically higher in the acacia samples in comparison to the other groups. Regarding sugar profile, the ratio between fructose and glucose was higher (*p*-value < 0.05) for the acacia samples in comparison to apple–dandelion and rhododendron honeys. This is due to a higher level of glucose in apple–dandelion honeys and a lower level of fructose in rhododendron honeys in comparison to the acacia ones. The fructose/glucose ratio has been reported to be a possible indicator for acacia honey in comparison to other monofloral honeys [5,33]. The volatile profile of acacia honey included most of the 24 investigated molecules except thymol and terpineol that were identified in only one sample and geranic acid, octane, γ-terpinene, and 2-nonanone which were absent in all the acacia samples. Except for the fructose/glucose ratio, no statistically significant compounds for the acacia floral origin were detected. This result could be due to the limited percentage of acacia pollens present in the samples in comparison to the abundance of other family groups (Table 4, Fagaceae, Rosaceae) which cannot determine a clear differentiation between this floral origin and others.

### 4.2. Apple–Dandelion Honeys

Melissopalynological analysis showed the presence of different classes of pollens in the apple–dandelion honey samples. The Rosaceae family, to which *Malus*/*Pyrus* belongs, was significantly more common in these samples in comparison to chestnut, honeydew, multifloral, and rhododendron honey, while the values were comparable to the acacia ones. In apple–dandelion honey, together with Rosaceae and Asteraceae, Vitaceae pollens were also present in an important amount. A wide presence of different pollen families was observed in the samples of this honey due to the botanical characteristic of the area and agricultural production. The apple–dandelion honey was produced between the Val di Non and Val d’Adige valleys (Trentino Alto-Adige) where there are simultaneous crops of apple trees (Rosaceae) and grapes (Vitaceae), along with spontaneous dandelion meadows (Asteraceae). Literature data, in agreement with our study, generally reports that in dandelion honey, the percentage of Asteraceae pollens is below 50%, and it is normally contaminated by different pollen species, such as Salicaceae [9,32].

As already highlighted [19], the levels of glucose were particularly high in apple–dandelion honey in comparison to the other samples (Figure 2a). In general, in many honey types, fructose is the dominant monosaccharide, but only in very few honey types, such as dandelion (*Taraxacum officinale*) and rape (*Brassica napus*) honey, the glucose component could be greater than the fructose one [21]. Thus, due to the high glucose content, this honey presents a high crystallization tendency if compared to other honey groups [32]. The fructose/glucose ratio in the apple–dandelion samples was significantly lower (*p*-value < 0.05) than in multifloral, acacia, rhododendron, and chestnut honeys due to the predominance of glucose (Figure 2a), indicating this parameter is effective to characterize this monofloral honey. Although the apple–dandelion honeys contained most of the investigated VOCs, unlike other floral honeys, benzeneacetaldehyde, thymol, geranic acid, octane, terpineol, γ-terpinene, and 2-nonanone were under the limit of detection; decanal and tetradecanoic acid were present only in one apple–dandelion sample. To the best of our knowledge, no studies have been published reporting the volatile organic composition of these honeys. Benzaldehyde, which was the most abundant compound in all the honey varieties, appeared to be particularly intense in the apple–dandelion samples, being statistically more common in this group (together with the acacia samples) with respect to the others (Figure 2b). Benzaldehyde has been reported as an important constituent of honeys of different floral origin, responsible for a pleasant aroma [22,24].

### 4.3. Rhododendron Honeys

Melissopalynological analysis showed the prevalence of Ericaceae pollens, which were significantly more common in the rhododendron honeys than in the other groups. With respect to the sugar profile, the levels of erlose were particularly high in rhododendron honey (Figure 3b), as already reported [19]. Erlose is known to be produced from sucrose by transglycosylation [42]; indeed, excluding the multifloral samples in which pollen contribution varied, rhododendron had the highest mean value of sucrose, indicating a relationship between the levels of these two sugars and the nectar obtained from these plants. The correlation between erlose and Ericaceae pollens was 0.72 while the correlation coefficient between erlose and sucrose was 0.48. With respect to VOCs, the rhododendron samples contained most compounds, excluding γ-terpinene and 2-nonanone which were always under the limit of detection. Terpineol appeared to be significantly more common in these samples (Figure 3b) in comparison to the other blossom honeys. This compound showed a correlation coefficient of 0.59 with erlose. Terpineol is considered a minor compound in honey despite its presence has been detected in many types of honey [43]. Terpineol has been found as a volatile component of leaves of *Rhododendron* species [44].

### 4.4. Honeydew Honeys

Honeydew honeys are expected to present a slightly different chemical profile from the other groups due to the different way of production of this kind of honey. Melissopalynological analysis is not applicable to this group because honeydew honey is not produced from the nectar of flowers [3]. The carbohydrate profile showed a major difference with respect to blossom honey. Particularly, raffinose and melezitose were detected at higher levels (Figure 4a,b) in comparison to the other honey types; these two compounds are known to characterize honeydew honey [45,46] and the presence of these sugars in blossom honey has been hypothesized to derive from contamination with honeydew honey [20]. Melezitose in honeydew honey can be present at a relatively high concentration, up to 30% of the total sugar content [47,48]. Raffinose has been identified as a component of honeydew produced by different species [49]. The sum of glucose and fructose was significantly higher in honeydew honey (*p*-value < 0.05) than in all the other honeys, as already reported [21]. With respect to the VOCs profile, honeydew honey showed the presence of most of the compounds in at least one sample, except for γ-terpinene. The most abundant compounds were furfural, octanoic acid, benzeneacetaldehyde, n-decanoic acid; however, anyone wasstatistically significant in honeydew honey in comparison to blossom honey.

### 4.5. Chestnut Honeys

Melissopalynological analysis showed the prevalence of Fagaceae pollens, which were significantly more common in chestnut honey than in the other groups. *Castanea* pollen is often strongly overrepresented in monofloral and many multifloral honeys [32]. Regarding the sugar profile, the ratio between fructose and glucose was higher (*p*-value < 0.05) for the chestnut samples in comparison to honeydew, apple–dandelion, and rhododendron honeys. Chestnut honey is known to contain high fructose and low glucose contents, determining a typically high value of the fructose and glucose ratio [32]. Therefore, the concentrations of these monosaccharides have been advanced as useful indicators for the classification of monofloral chestnut honey [5]. The sugar nigerose appeared to be significantly more common in chestnut honey in comparison to the other floral groups (Figure 5b). Nonanoic acid appeared to be a significant VOC in chestnut honey, excluding honeydew in which the mean abundance was statistically similar (Figure 5a); this result agrees with the previous results [50,51] which demonstrated nonanoic acid presence in chestnut flowers as well [52]. Several compounds belonging to volatile organic components of honey can originate from heating, storage, microbiological transformations, as well as from environmental contamination [25]. Exogenous substances such as furfural and its derivatives (i.e., 5-methylfurfural, furfuryl alcohol, 5-hydroxymethylfurfural, furfuryl-n-butyrate) were found in various honey types, as previously described [25,29,50,53,54,55]. These compounds are related to heating practices and preservation conditions of honey; although legislation provides the maximum levels in honey only for 5-hydroxymethyl-2-furaldehyde (HMF), these compounds normally cannot indicate good honey quality. In our chestnut honey, a significant level of furfural was observed (Figure 5c) in comparison to the other kinds of honey. Previous studies [56] reporting the concentration of furanic compounds in various monofloral and multifloral honeys indicated the presence of furfural only in chestnut honey, hypothesizing, differently from HMF, a botanical origin for this compound.

### 4.6. Multifloral Honeys

As expected, melissopalynological analysis showed a very complex composition of pollens in the multifloral honey samples (Figure 6). Most of the samples showed the presence of high percentages (>50%) of Ericaceae or Fagaceae pollens accompanied by lower percentages of Fabaceae (3–23%) or Rosaceae (4–25%), the latter being the most frequent in the multifloral samples (*n* = 17). The presence of Asteraceae (3–12%) and Salicaceae (3–37%) pollens was also observed, together with Ranunculaceae (3–8%), Scrophulariaceae (3–23%), and Rubiaceae (3–8%). Apiaceae and Tiliaceae pollens were also observed in five and four samples, respectively. Regarding the chemical profile, no compounds appeared to be characteristic of this group due to the heterogeneity of these samples which determines a very different chemical composition with high intragroup standard deviations for most of the compounds.

To try the identification of particular substances related to the pollen composition, all the samples (multifloral and monofloral) were subdivided in four groups depending on the prevalent class of the present pollens: Fagaceae group with a Fagaceae pollen frequency > 25% (AD45, AD43, M47-D, M47-P, M47-C, R2-18, C13-18, M44, C12-18, C18-18, C52-D, C52-P, C52-C, M23-18, A1-18, M7-18, M11-18, M9-18, C40, M6-18, M5-18); Ericaceae group with an Ericaceae pollen frequency > 25% (M36, M3-18, M16-18, R27-18, R17-18, R18-18, M21-18, M20-18, R14-18, R24-18, R4-18, M38, M37, M19-18, M42, M41, M39); Rosaceae group with a Rosaceae pollen frequency > 25% (AD-25, M22-18, A28-18). The last group included the samples in which no pollen family reached at least the frequency of 25% (f < 25 group) (M46-C, M46-P, M46-D). Statistical analysis was conducted on the four honey sample groups, considering only chemical data (sugars and VOCs), showing some interesting results. Particularly, fructose, erlose, and benzaldehyde showed significantly different values in the four groups (Figure 7a–c). Fructose demonstrated higher values in the group with the family pollen frequency lower than 25% than in the other groups (Figure 7a). This group had a contribution of various pollen families, particularly of Fagaceae, Fabaceae, Rosaceae, and Scrophulariaceae, including numerous herbaceous species, which could determine higher levels of fructose. Erlose (Figure 7b) appeared to be significantly more common in the Ericaceae group, in agreement with what had already been highlighted for the rhododendron samples (Section 4.3), confirming the relationship of this compound with the pollen family of Ericaceae. Benzaldehyde (Figure 7c) showed a higher abundance in the Rosaceae group than in the others; this compound has, indeed, a correlation coefficient of 0.72 with the variable Rosaceae. Benzaldehyde is known to be abundant in seeds of Rosaceae family plants, especially in *Prunus dulcis* sp.; in honey, it is generally associated with fresh honey odor, and it has been shown to diminish during honey storage [22,57]. Interestingly, with this new classification, the role of benzaldehyde is clarified in comparison to the monofloral classification in which this compound appeared to be significant in the apple–dandelion samples. The classification based on the pollen family appeared effective in this case.

### 4.7. Overall Findings

To test the effectiveness of the classification based on the prevalent pollen family, a PLS-DA was performed (Figure 8a–c). The model was built using 44 blossom honey samples, excluding the honeydew ones, and 38 variables (organic compounds). Melissopalynological results were excluded in order to assess the relationship between the pollen frequencies and the chemical composition of honey. The model was validated using the leave-one-out cross-validation (LOOCV) method. The cross-validated correlation coefficient (Figure 8c) indicated the best predictive ability for a three-component model (Q2, 0.45) with an accuracy of 0.75 and a correlation coefficient of 0.81 (R2). The three-dimensional score plot permitted the separation of the Ericaceae, Fagaceae, and f < 25 groups (Figure 8a), explaining, however, only 35.1% of the variance. The most relevant variables for the model (VIP > 1, variable importance in projection) were erlose, benzaldehyde, hotrienol, terpineol, octanoic acid, benzeneacetaldehyde, raffinose, fructose, linalool, kojibiose, and furfural (Figure 8b) The terpenic derivatives hotrienol, terpineol, linalool, and the sugar fructose showed higher levels in the f < 25 group. Linalool and terpineol are known to be present in many kinds of honey and are associated with a sweet fresh flowery scent [43]. Hotrienol is known to be a thermally generated product, but there are findings that support its natural occurrence in non-thermally treated honey [58]. Hotrienol could also derive from linalool [59]. The possible explanation for high fructose levels in this group was previously discussed in Section 4.6. The trisaccharides erlose and raffinose were significant for the model, being higher in the Ericaceae group than in the other blossom honeys, together with benzene acetaldehyde and octanoic acid. Benzene acetaldehyde has been identified in many kinds of honeys, including in monofloral honeys of the Ericaceae family [60], and it is known to possess a pleasant honey-like aroma [25]. Octanoic acid has already been identified in monofloral Ericaceae honey [61,62]. Furfural was significant for the model, with high levels in the Fagaceae group, confirming what was previously observed for chestnut honey (Section 4.5). The disaccharide kojibiose was significant for the model, with high values in the Rosaceae group, together with benzaldehyde, as already discussed previously in Section 4.6.

These results indicate that there is a relationship between honey botanical origin and chemical composition, highlighting the inadequacy of the employed monofloral/multifloral honey classification. This ineffectiveness is mainly due to the inhomogeneity of the pollens present among samples belonging to the same “botanical class”, leading to different chemical compositions. Conversely, samples classified in different botanical classes showed similarity from the melissopalynological point of view and, therefore, a similar chemical pattern. The botanical classification of monofloral honeys, indeed, does not reflect the typology and abundance of the pollens contained in honeys due to the underrepresentation of some classes of pollens (e.g., acacia, apple–dandelion) and the generic multifloral classification. Particularly, multifloral honeys, based on sugars and the VOCs content, could have organoleptic properties similar to those of monofloral honeys, such as rhododendron and chestnut honeys, which commercially are much more valuable. Emblematic is the case of chestnut honey, which in many cases showed *Castanea* pollen percentages very close to some multifloral honeys of the Fagaceae class (90–98% and 73–81%, respectively) (Figure 6) and demonstrating, therefore, a chemical profile more similar to those of chestnut honeys than to those of other multifloral honeys. The PLS-DA model appeared to be capable of partially illustrating the relationship between the honey chemical profile and the prevalent pollen families, improving the classification of samples, especially of the multifloral ones. The low variance was probably due to the simplified sample grouping based only on one prevalent pollen family which, in half of the cases, did not exceed the 50% frequency.

## 5. Conclusions

The purpose of this study was to investigate the chemical composition of honeys produced in one specific region of northern Italy from different botanical species. Melissopalynological analysis was applied to accurately ascertain the botanical origin of the investigated honey samples. Chemical characterization was carried out considering the carbohydrate content and the volatile organic compounds. The results show that the chemical composition of honey is related to its botanical origin and some parameters investigated in this paper, permitting the differentiation of the five botanical origin honeys studied. Particularly, the fructose/glucose ratio appeared to be particularly high in the acacia samples; high levels of glucose and benzaldehyde were characteristic of apple–dandelion honey. Erlose and terpineol were significantly intense in rhododendron honey while raffinose and melezitose appeared to be markers of honeydew honey. Finally, nigerose, nonanoic acid, and furfural were significantly common in chestnut honeys. Multifloral honeys, however, had a heterogeneous chemical composition in relation to the different presence and abundance of pollens. To improve the classification of samples and test the relationship between pollens and the chemical profile of honey, all the blossom samples were subdivided on the basis of the prevalent pollen family; the relationship was investigated using a PLS-DA model which appeared to be effective in the separation of the samples and identifying common chemical characteristics among the groups. Particularly, the Ericaceae honeys (including multifloral and rhododendron honeys) were characterized by high levels of erlose, raffinose, octanoic acid, and benzeneacetaldehyde while the Rosaceae group, which included the acacia and apple–dandelion samples, was characterized by high levels of benzaldehyde and kojibiose. The f < 25 group, with all pollen family frequencies lower than 25%, showed high levels of fructose, linalool, hotrienol, and terpineol. The Fagaceae honeys (including multifloral, apple–dandelion, rhododendron, acacia, and chestnut samples) showed the presence of high levels of furfural. The results of this study highlight the weakness of the applied honey classification, both in the case of monofloral honeys, in which the characteristic pollens are often underrepresented, and of multifloral honeys, in which the pollen composition can vary a lot. The typology and abundance of pollen influence the chemical composition of honeys and, therefore, their organoleptic properties, which definitely should determine the effective commercial value.

These results are based only on a limited set of samples and should be considered preliminary and relative only to the honeys collected in the studied area; our results need to be confirmed by samples collected in further years of production, in order to verify the hypothesis elaborated here. However, our study offers an interesting new point of view on the classification of honey and highlights the importance of considering the real chemical composition of honey.

## Figures and Tables

**Figure 1 foods-11-02441-f001:**
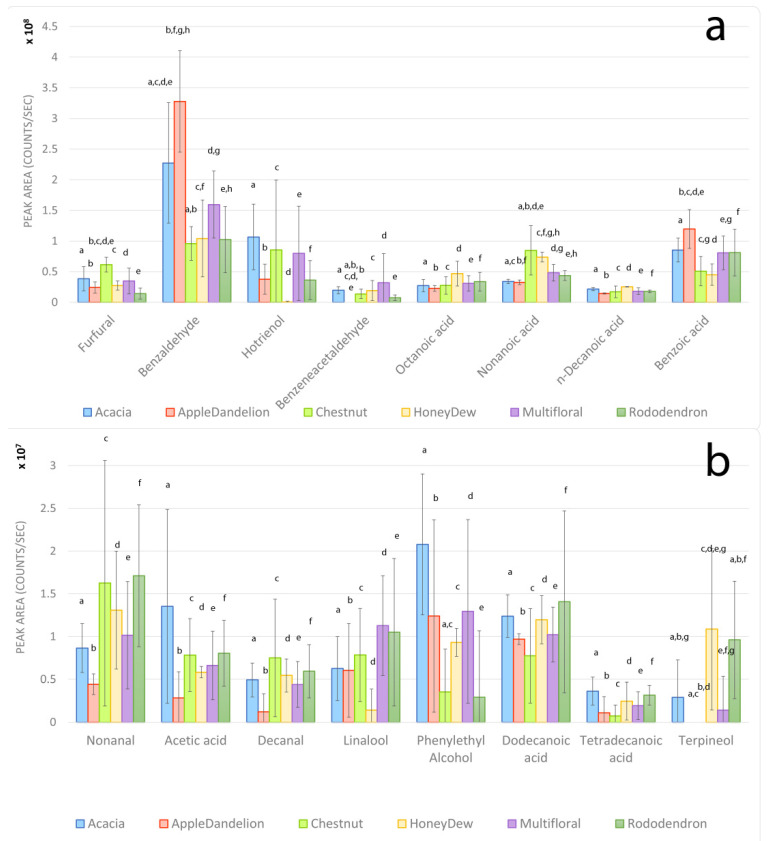
Histograms of more intense (**a**) and less intense (**b**) VOCs in the honey samples. Error bars represent the standard deviations. Same letters indicate statistically significant differences.

**Figure 2 foods-11-02441-f002:**
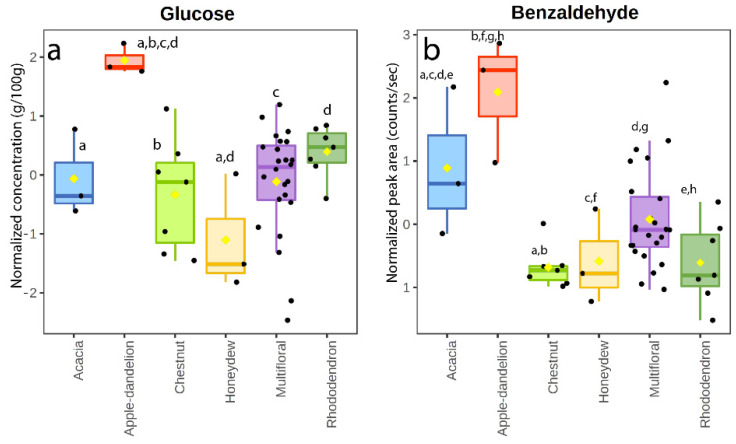
Box and whisker plot of glucose (**a**) and benzaldehyde (**b**) in the honey samples. Same letters indicate statistically significant differences.

**Figure 3 foods-11-02441-f003:**
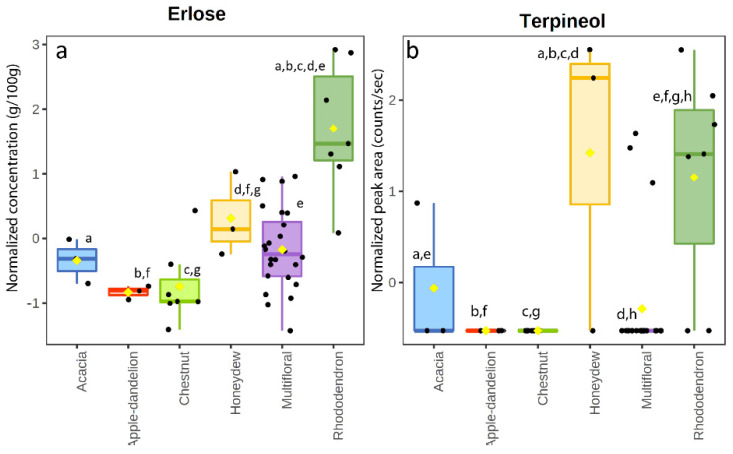
Box and whisker plot of erlose (**a**) and terpineol (**b**) in the honey samples. Same letters indicate statistically significant differences.

**Figure 4 foods-11-02441-f004:**
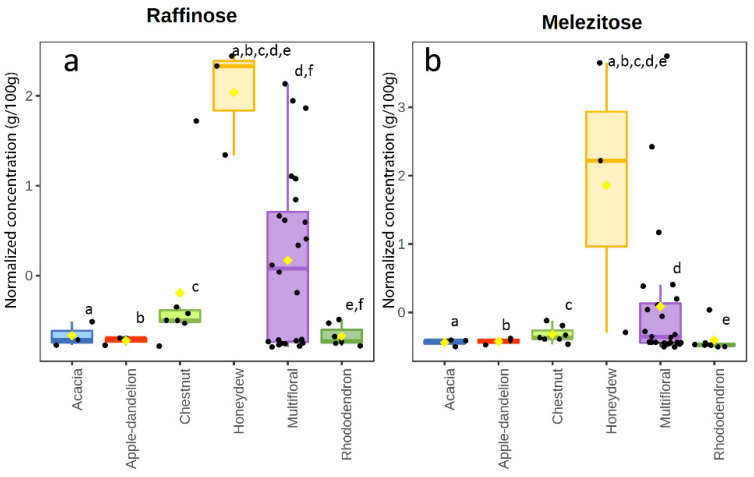
Box and whisker plot of raffinose (**a**) and melezitose (**b**) in the honey samples. Same letters indicate statistically significant differences.

**Figure 5 foods-11-02441-f005:**
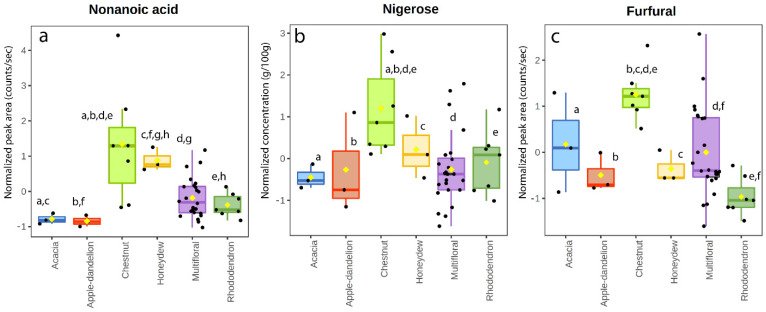
Box and whisker plot of nonanoic acid (**a**), nigerose (**b**), and furfural (**c**) in the honey samples. Same letters indicate statistically significant differences.

**Figure 6 foods-11-02441-f006:**
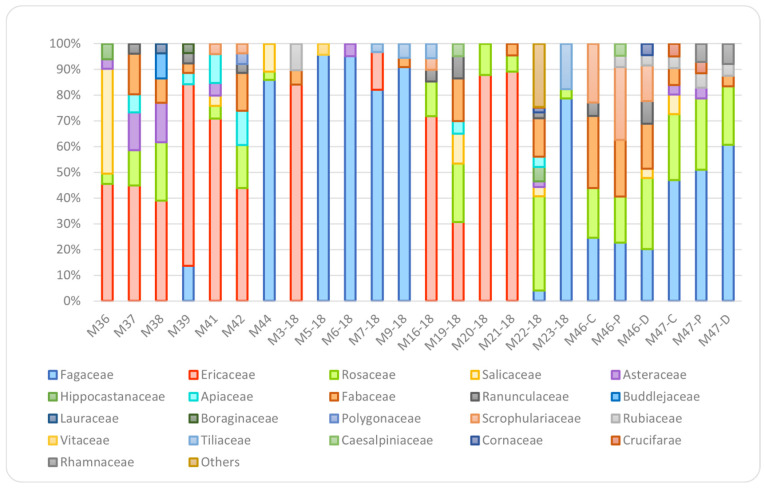
Relative abundancies of pollen families in the multifloral honey samples.

**Figure 7 foods-11-02441-f007:**
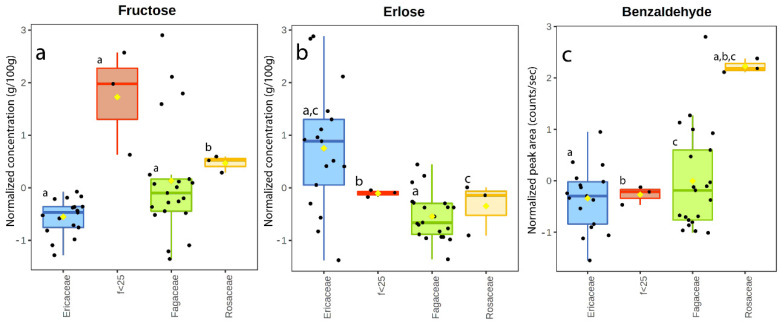
Box and whisker plot of fructose (**a**), erlose (**b**), and benzaldehyde (**c**) in the honey samples. Same letters indicate statistically significant differences.

**Figure 8 foods-11-02441-f008:**
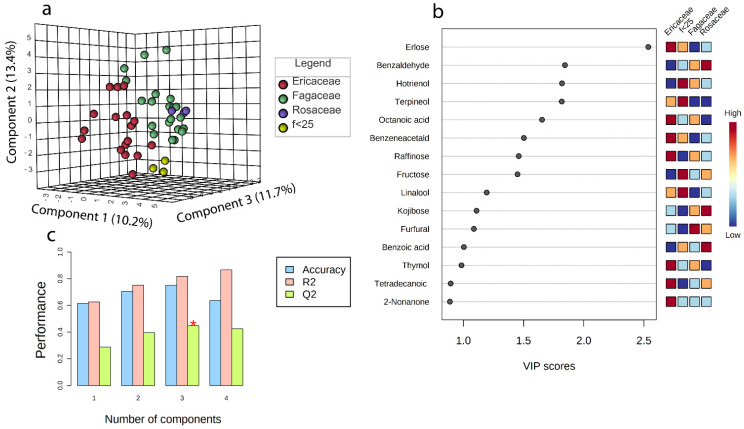
Three-dimensional score plot (**a**); VIP score (**b**); and performance parameter histogram (the red asterisk indicates the higher R2 value in relation to the component number) (**c**) of the PLS-DA model.

**Table 1 foods-11-02441-t001:** Minimum percentage of pollen required for the characterization of monofloral honey in some European countries ^1^.

Pollen Grain	Honey Type	Italy	Germany	Greece	Croatia	Serbia
*Robinia pseudoacacia*	Locust or acacia	15	20		20	20
*Citrus* spp.	Orange	10	20	3	10	
*Taraxacum* spp.	Dandelion	5	15			20
*Lavandula* spp.	Lavander		15		10	
*Rosmarius officinalis*	Rosemary		20			20
*Thymus* spp.	Thyme	15	20	18		
*Tilia* spp.	Linden or lime		20		25	25
*Castanea sativa*	Chestnut	90	90	87	85	85
*Rhododendron*	Snowrose or alpenrose	25				

^1^ Beckh and Camps, 2009 [8]; Thrasyvoulou et al., 2018 [4]; Persano Oddo et al., 1995 [9]; Italian legislation UNI 11382:2010 [11].

**Table 2 foods-11-02441-t002:** Descriptive characteristics of the honey samples analyzed with different botanical and geographical origin and harvest year.

Multifloral Honey				Monofloral and Honeydew Honey			
Sample Code	Botanical Origin	Geographical Origin	Harvest Year	Sample Code	Botanical Origin	Geographical Origin	Harvest Year
M36	Multifloral	Val di Fiemme	2017	A1-18	Acacia	Valsugana	2018
M37	Multifloral	Val di Fiemme	2017	A11-18	Acacia	Valsugana	2018
M38	Multifloral	Val di Fiemme	2017	A28-18	Acacia	Val di Non	2018
M39	Multifloral	Val di Fiemme	2017	AD43	Apple–dandelion	Val di Non	2018
M41	Multifloral	Val di Cembra	2017	AD45	Apple–dandelion	Val d’Adige	2017
M42	Multifloral	Val di Fassa	2017	AD25-18	Apple–dandelion	Val di Non	2018
M44	Multifloral	Val di Fiemme	2017	R2-18	Rhododendron	Valsugana	2018
H3-18	Multifloral	Val di Non	2018	R4-18	Rhododendron	Val di Non	2018
M5-18	Multifloral	Val di Non	2018	R14-18	Rhododendron	Val di Fiemme	2018
M6-18	Multifloral	Val di Non	2018	R17-18	Rhododendron	Val di Fiemme	2018
M7-18	Multifloral	Valsugana	2018	R18-18	Rhododendron	Val di Fiemme	2018
M9-18	Multifloral	Valsugana	2018	R24-18	Rhododendron	Valsugana	2018
M16-18	Multifloral	Val di Fiemme	2018	R27-18	Rhododendron	Val di Non	2018
M19-18	Multifloral	Val di Fiemme	2018	HD15-18	Honeydew	Val di Fiemme	2018
M20-18	Multifloral	Val di Fiemme	2018	HD26-18	Honeydew	Val di Non	2018
M21-18	Multifloral	Val di Fiemme	2018	HD29-18	Honeydew	Val di Non	2018
M22-18	Multifloral	Val d’Adige	2018	C8-18	Chestnut	Valsugana	2018
M23-18	Multifloral	Valsugana	2018	C12-18	Chestnut	Valsugana	2018
M46-C	Multifloral	Val d’Adige	2018	C13-18	Chestnut	Valsugana	2018
M46-D	Multifloral	Val d’Adige	2018	C40	Chestnut	Val di Fiemme	2017
M46-P	Multifloral	Val d’Adige	2018	C52-C	Chestnut	Val d’Adige	2018
M47-C	Multifloral	Val d’Adige	2018	C52-D	Chestnut	Val d’Adige	2018
M47-D	Multifloral	Val d’Adige	2018	C52-P	Chestnut	Val d’Adige	2018
M47-P	Multifloral	Val d’Adige	2018				

**Table 3 foods-11-02441-t003:** Results of melissopalynological analysis of honeydew honeys: HDE/P and pollen grains (PG) per gram for each sample.

Sample	Floral Type	Honeydew Elements (HDE/P)	PG/g
HD15-18	Honeydew	3.13	19,602
HD26-18	Honeydew	3.57	35,281
HD29-18	Honeydew	7.61	15,964

**Table 4 foods-11-02441-t004:** Minimum and maximum percentages of prevalent pollens in monofloral honeys according to the family classification and the genus marker. Same letters indicate statistically significant differences (*p*-value < 0.05).

Family Classification							Genus Classification			
Floral origin	Fagaceae	Ericaceae	Rosaceae	Salicaceae	Asteraceae	Fabaceae	*Malus*/*Pyrus* (Apple–Dandelion)	Asteraceae, T-Form (Apple–Dandelion)	*Robinia Pseudoacacia* (Acacia)	*Castanea* (Chestnut)
Chestnut	90–98 ^a,b,c^	0–4 ^a^	0 ^a,c^	0 ^a^	0 ^a^	0 ^a^	0 ^a,b^	0 ^a^	0 ^a^	90–98 ^a,b,c^
Rhododendron	0–64 ^a^	31–90 ^a,b,c^	0–5 ^b,d^	0 ^b^	0 ^b^	0 ^b^	0 ^c,d^	0 ^b^	0 ^b^	0–64 ^a^
Acacia	14–65 ^b^	0–31 ^b^	15–38 ^a,b^	0 ^c^	0 ^c^	17–20 ^a,b^	0–19 ^a,b,c^	0 ^c^	17–19 ^a,b,c^	0–65 ^b^
Apple–dandelion	0–49 ^c^	0 ^c^	12–38 ^c,d^	0–19 ^d^	0–23 ^a,b,c^	0–11 ^a,b^	12–22 ^b,d^	2–23 ^a,b,c^	0 ^c^	0–49 ^c^

**Table 5 foods-11-02441-t005:** Concentrations of carbohydrates, mean values and RSD (in brackets), in the 47 honey samples, of which 33 were previously reported [19]. Data are expressed in g/100 g^−1^. Same superscript letters (for each compound) indicate statistically significant differences. Glu: glucose; Fru: fructose; Sucr: sucrose; Meli: melizitose; Lacto: lactose; Lactu: lactulose; Koji: kojibiose; Tura: turanose; Pala: palatinose; Nige: nigerose; Mele: melezitose; Raffi: raffinose; Isomal: isomaltose; Erlo: erlose.

Sample	Floral Type	Glu	Fru	Fru/Glu	Sucr	Meli	Lacto	Lactu	Koji	Tura	Pala	Nige	Mele	Raffi	Isomal	Erlo
M36	Multifloral	19.1	43.4	2.27	0.02	0.054	4.12	1.85	0.63	3.41	4.01	0.89	0.09	0.03	0.36	0.08
M37 ^§^	Multifloral	18.6	40.7	2.19	0.08	0.093	2.83	0.95	0.45	2.00	0.83	0.65	2.69	0.59	0.20	0.91
M38 ^§^	Multifloral	17.1	39.9	2.33	0.11	0.068	2.65	1.01	0.77	2.35	1.21	0.73	2.11	0.78	0.14	1.31
M39 ^§^	Multifloral	17.2	37.7	2.19	0.29	0.077	2.10	0.84	0.48	2.23	0.78	0.63	1.74	1.13	0.08	3.53
M41 ^§^	Multifloral	15.3	36.1	2.36	0.23	0.065	2.28	0.82	0.50	1.78	0.69	0.53	3.47	1.21	0.13	2.26
M42	Multifloral	15.9	38.5	2.42	0.24	0.108	2.77	0.82	0.68	2.40	0.96	0.62	3.38	1.10	0.17	2.79
M44 ^§^	Multifloral	18.6	44.3	2.38	0.10	0.048	3.59	1.00	1.21	2.38	1.96	0.82	2.36	0.61	0.16	1.60
M3-18 ^§^	Multifloral	17.4	43.5	2.50	0.20	0.036	2.31	0.81	1.21	1.34	0.31	0.86	6.31	0.38	0.09	1.72
M5-18 ^§^	Multifloral	13.6	37.6	2.77	0.13	0.038	1.61	0.59	0.55	0.96	0.29	0.52	10.96	0.58	0.08	1.34
M6-18 ^§^	Multifloral	12.9	35.6	2.76	0.11	0.084	2.23	0.78	0.73	0.83	0.39	0.59	15.89	0.79	0.10	0.68
M7-18 ^§^	Multifloral	17.9	43.5	2.44	1.09	0.034	1.65	0.56	0.64	1.13	0.23	0.55	0.23	0.35	0.09	2.52
M9-18 ^§^	Multifloral	16.2	44.2	2.72	0.15	0.017	1.45	0.61	0.55	1.20	0.35	0.56	0.60	0.03	0.08	1.14
M16-18 ^§^	Multifloral	19.3	42.9	2.23	0.43	0.038	2.22	0.71	0.76	1.70	0.63	0.58	0.74	0.68	0.10	2.96
M19-18 ^§^	Multifloral	17.7	40.4	2.28	0.37	0.050	1.76	0.53	0.57	0.87	0.28	0.47	0.31	0.50	0.07	2.80
M20-18 ^§^	Multifloral	19.5	44.9	2.30	0.64	0.032	1.82	0.71	0.68	1.97	0.52	0.60	0.34	0.26	0.10	3.57
M21-18	Multifloral	20.6	45.1	2.19	0.60	0.048	2.02	0.75	0.52	1.70	0.56	0.58	0.64	0.48	0.11	3.64
M22-18 ^§^	Multifloral	19.0	51.2	2.69	0.60	0.036	1.74	0.58	0.64	1.77	0.67	0.53	0.07	0.01	0.08	1.96
M23-18 ^§^	Multifloral	20.1	45.6	2.26	0.12	0.032	2.11	0.67	0.73	1.56	0.35	0.65	0.92	0.05	0.09	0.83
M46-C ^§^	Multifloral	18.0	51.5	2.86	4.03	0.031	1.26	0.43	0.41	1.32	0.44	0.41	0.24	0.02	0.08	2.03
M46-P	Multifloral	18.3	67.1	3.67	4.65	0.020	1.28	0.45	0.43	1.31	0.34	0.45	0.31	0.02	0.04	2.10
M46-D	Multifloral	18.4	62.3	3.38	3.77	0.023	1.20	0.43	0.00	1.19	0.00	0.44	0.34	0.03	0.04	1.91
M47-C ^§^	Multifloral	19.6	48.4	2.47	2.30	0.028	1.67	0.53	0.65	1.86	0.88	0.58	0.33	0.04	0.08	1.72
M47-P	Multifloral	18.6	59.2	3.19	2.04	0.027	1.55	0.51	0.51	1.51	0.82	0.55	0.29	0.04	0.05	1.77
M47-D	Multifloral	19.3	60.8	3.15	2.05	0.025	1.68	0.55	0.45	1.61	0.37	0.56	0.33	0.04	0.05	1.60
	Mean (SD)	17.8 ^c^ (1.9)	46.0 ^a^ (8.6)	2.58 ^a,b^ (0.41)	1.01 ^a^ (1.38)	0.046 ^a^ (0.024)	2.08 ^a^ (0.71)	0.73 ^a^ (0.29)	0.61 ^a^ (0.24)	1.68 ^a^ (0.59)	0.74 ^a^ (0.80)	0.60 ^d^ (0.12)	2.28 ^d^ (3.81)	0.41 ^d,f^ (0.39)	0.11 ^a^ (0.07)	1.95 ^e^ (0.94)
A1-18 ^§^	Acacia	16.8	44.8	2.67	0.29	0.031	1.89	0.65	0.63	1.27	0.35	0.61	0.41	0.13	0.09	1.74
A11-18 ^§^	Acacia	17.3	46.5	2.68	0.13	0.021	1.56	0.57	0.61	0.88	0.36	0.54	0.10	0.02	0.07	1.17
A28-18 ^§^	Acacia	19.7	48.7	2.47	0.33	0.028	1.50	0.69	0.77	1.56	0.61	0.56	0.44	0.04	0.05	2.19
	Mean (SD)	18.0 ^a^ (1.6)	46.7 ^b^ (2.0)	2.61 ^c,d^ (0.12)	0.25 ^b^ (0.11)	0.027 ^b^ (0.005)	1.65 ^b^ (0.21)	0.64 ^b^ (0.06)	0.67 ^b^ (0.09)	1.24 ^b^ (0.34)	0.44 ^b^ (0.15)	0.57 ^a^ (0.04)	0.32 ^a^ (0.19)	0.06 ^a^ (0.06)	0.07 ^b^ (0.02)	1.70 ^a^ (0.51)
AD25-18 ^§^	Apple–dandelion	22.8	50.6	2.22	0.09	0.041	2.87	1.02	0.94	2.67	1.81	0.79	0.54	0.05	0.13	0.79
AD43 ^§^	Apple–dandelion	21.9	47.0	2.14	2.90	0.032	2.33	0.74	0.72	1.99	0.85	0.47	0.20	0.02	0.12	1.00
AD45 ^§^	Apple–dandelion	21.8	47.2	2.16	0.12	0.027	1.78	0.63	0.56	1.94	1.04	0.53	0.41	0.05	0.13	1.10
	Mean (SD)	22.2 ^a,b,c,d^ (0.5)	48.3 ^c^ (2.0)	2.18 ^a,c,d,e^ (0.04)	1.04 ^c^ (1.62)	0.033 ^c^ (0.007)	2.33 ^c^ (0.55)	0.80 ^c^ (0.20)	0.74 ^c^ (0.19)	2.20 ^c^ (0.40)	1.23 ^c^ (0.51)	0.60 ^b^ (0.17)	0.38 ^b^ (0.17)	0.04 ^b^ (0.02)	0.13 ^c^ (0.01)	0.96 ^b,f^ (0.16)
R2-18 ^§^	Rhododendron	17.2	42.9	2.48	0.29	0.021	2.14	0.73	0.97	1.31	0.34	0.80	2.09	0.14	0.08	2.34
R4-18 ^§^	Rhododendron	18.4	41.7	2.27	1.19	0.022	1.76	0.76	0.81	1.61	0.14	0.67	0.18	0.06	0.08	4.40
R14-18 ^§^	Rhododendron	19.4	42.7	2.20	0.84	0.031	1.84	0.72	0.64	1.91	0.41	0.68	0.18	0.12	0.09	4.16
R17-18 ^§^	Rhododendron	19.7	44.7	2.27	5.54	0.017	1.56	0.83	0.48	2.51	0.39	0.54	0.21	0.04	0.00	6.57
R18-18 ^§^	Rhododendron	18.7	42.2	2.26	3.45	0.020	1.37	0.65	0.47	1.46	0.15	0.53	0.09	0.03	0.07	5.40
R24-18 ^§^	Rhododendron	19.9	45.8	2.31	0.85	0.036	1.81	0.57	0.50	1.50	0.25	0.49	0.10	0.03	0.11	3.87
R27-18 ^§^	Rhododendron	19.1	43.5	2.28	2.66	0.010	1.63	0.88	0.59	1.89	0.00	0.65	0.29	0.02	0.07	6.50
	Mean (SD)	18.9 ^d^ (0.9)	43.4 ^d^ (1.4)	2.30 ^b,d,e,f^ (0.09)	2.12 ^d^ (1.88)	0.023 ^d^ (0.009)	1.73 ^d^ (0.24)	0.73 ^d^ (0.11)	0.64 ^d^ (0.19)	1.74 ^d^ (0.40)	0.24 ^d^ (0.15)	0.62 ^e^ (0.11)	0.45 ^e^ (0.73)	0.06 ^e,f^ (0.05)	0.07 ^d^ (0.03)	4.75 ^a,b,c,d,e^ (1.52)
HD15-18 ^§^	Honeydew	18.1	39.2	2.16	0.25	0.049	2.22	0.66	0.55	1.26	0.62	0.57	0.87	0.89	0.08	2.42
HD26-18 ^§^	Honeydew	14.9	35.7	2.40	0.95	0.033	2.88	0.70	0.72	1.49	0.73	0.65	10.21	1.34	0.10	3.75
HD29-18 ^§^	Honeydew	14.3	36.1	2.53	0.29	0.044	2.73	0.88	0.80	1.68	0.60	0.78	15.52	1.29	0.11	1.85
	Mean (SD)	15.5 ^a,d^ (1.8)	37.5 ^e^ (1.8)	2.44 ^g^ (0.21)	0.41 ^e^ (0.36)	0.043 ^e^ (0.007)	2.41 ^e^ (0.50)	0.74 ^e^ (0.10)	0.68 ^e^ (0.11)	1.62 ^e^ (0.34)	0.72 ^e^ (0.15)	0.66 ^c^ (0.09)	8.82 ^a,b,c,d,e^ (6.06)	1.02 ^a,b,c,d,e^ (0.37)	0.09 ^e^ (0.02)	2.31 ^d,f,g^ (1.07)
C40	Chestnut	15.3	36.7	2.41	0.25	0.088	2.90	0.95	0.68	2.74	1.12	0.76	1.24	1.04	0.10	2.85
C8-18	Chestnut	15.0	42.3	2.81	0.05	0.056	3.87	1.36	0.97	2.65	0.96	1.05	0.23	0.02	0.21	0.10
C12-18	Chestnut	16.1	42.6	2.65	0.10	0.055	2.71	0.79	0.76	1.62	0.65	0.68	0.71	0.19	0.12	0.71
C13-18	Chestnut	18.8	47.8	2.53	0.21	0.042	2.64	0.94	1.17	2.37	0.77	0.99	1.51	0.12	0.10	1.61
C52-C	Chestnut	17.8	47.4	2.66	0.28	0.032	1.98	0.70	0.72	1.68	0.60	0.68	0.51	0.13	0.09	0.75
C52-P	Chestnut	18.2	63.4	3.48	0.24	0.033	1.90	0.69	0.55	1.71	0.55	0.65	0.52	0.13	0.05	0.75
C52-D	Chestnut	20.4	69.7	3.41	0.28	0.038	2.23	0.79	0.78	1.85	0.84	0.81	0.58	0.16	0.06	0.91
	Mean (SD)	17.4 ^b^ (2.0)	50.0 ^f^ (12.0)	2.85 ^e,f,g^ (0.43)	0.20 ^f^ (0.09)	0.049 ^f^ (0.019)	2.60 ^f^ (0.67)	0.89 ^f^ (0.23)	0.80 ^f^ (0.20)	2.09 ^f^ (0.48)	0.78 ^f^ (0.21)	0.80 ^b,c,d,e^ (0.16)	0.76 ^c^ (0.45)	0.26 ^c^ (0.35)	0.11 ^f^ (0.05)	1.10 ^c,g^ (0.89)

^§^ Previously reported data [19].

## Data Availability

The data presented in this study are available on request from the corresponding author.

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
