# Peer review of "Characterization of Botanical Origin of Italian Honey by Carbohydrate Composition and Volatile Organic Compounds (VOCs)"

_foods, 2022, doi:10.3390/foods11162441_

Round 1
Reviewer 1 Report
This paper describes the chemical characterization of honeys of different botanical origin (multifloral, acacia, apple-dandelion, rhododendron, honeydew, and chestnut), produced and collected by beekeepers in the Trentino Alto-Adige region (Italy). This characterization is based on: (1) Melissopalynological analysis in order to verify the botanical origin of samples and determine the frequency of different pollen families. (2) The carbohydrate composition (fourteen sugars). (3) The profile of VOCs in order to investigate the relation between pollen composition and the chemical profile of honey.
The manuscript is suitable for this journal. Although there are some noteworthy novelties in the work, such as the study of dandelion honey ( including its volatile organic composition), the most important part of the work is found in the conclusions. Thus, the authors highlight the weakness of the applied honey classification, both in the case of monofloral and of multifloral, which composition in pollens can vary a lot.
I suggest the authors to apply some further classification or prediction models, to enhance the work. There are some comments. They are presented below.
Materials and methods
Page 2 Line 103-104: Are they then honeys that have not undergone any filtering or centrifugation process or whatever, before being acquired for work? If so, authors should specify it in the text
Page 2 Line 107-108: The number of samples of some types of honey is not very representative. In fact, in some of the graphs that are shown throughout the article, it makes little sense to show the box&whisker diagram of some types of honey with only 3 samples collected for each of them. In my opinion, this aspect should be clarified in the text. In a first reading, this fact was for me a real problem when it came to "digest" the results. However, the importance of the conclusions of the work ends up minimizing this problem. However, I insist, the authors should clarify at this point why the number of samples of some varieties is so low.
Page 2 Line 111-112: How can you be sure of the representativeness of the acquired samples? Please comment it.
Some references are missing that should be included in sections 2.5.1 and 2.5.2, in order to support the suitability of the procedures described in them.
Results
Table 2. Why are the minimum and maximum percentages indicated and not also (or only) mean values with their standard deviation included?
In addition, the mean values together with the standard deviation should include an indication (for example, a superscript letter) meaning the significant differences between honey types.
Table 3. Please, include the standard deviation. It is not clear whether or not the mean values included at the end of the table already take into account the new values measured in this article. If the answer is no, they should have it. In addition, and as in Table 2, the mean values together with the standard deviation should include an indication (for example, a superscript letter) meaning the significant differences between honey types.
Discussion
Please remove the letters a, b and/or c in the bold title of figures 2, 3, 4, 6, and 7.
Minor comments
I suggest a detailed revision of the text, to adjust some minor typos and sentences. Just mentioning a few:
Page 1 Line 60, Page 3 Line 98: …influences…
Page 17 Line 602-605: The results show that the chemical composition of honey is related to its botanical origin and that some parameters investigated in this paper permit the differentiation of the five botanical origin honeys studied.
Reviewer 2 Report
In this manuscript, the botanical origin of Italian honey were characterized by melissopalynological analysis, sugur composition and VOCs. Overall, the experimental design and data analysis is resonable. However, there are still many written problems in the manuscript, which should be carefully revised and examined.
For example, the ambiguous meaning of some long sentences may cause misunderstanding for the readers, such as "The data were.." in Line 95-100.
The references isn't specific in "[20-27]" in Line 93.
The descriptin of "a temperature lower than 40°C" in Line 118 should be more specifc.
Place or country should be added to " Sigma Aldrich" in Line 121 and 125.
Moreover, please check "melissopalinological" and "melissopalynological" in the manuscript. Please check "400 X e 600-1000 X" in Line 140.
"(RSD%)" in Line 211 and "(PLSA-DA)" in Line 545 should be deleted and please check the all text.
In Figure 1, 2, 3 and 4, carbohydrates profile are showed with box and whisker plot separately. These individual figures make information to be fragmented. So it is recommanded to integrate them into one figure or table.
Reviewer 3 Report
In the current manuscript, the chemical characterization of kinds of honey of different botanical origins from Italy was investigated. The structure and the results are well written and then discussed. In my opinion, the current manuscript needs some corrections and appropriate responses to the following comments:
L28: Use other keywords that are not in the title of the article. Remove VOCs and botanical origin.
L108: Why did you put Table S1 in Supplementary Materials? This Table, which shows the complete characteristics of the honey samples examined in this research, should be visible to everyone in the text of the manuscript.
L135: Did you use a refrigerated centrifuge? Write the device specifications.
L163: What ratio were the samples diluted?
L178: What is the reason for adding NaCl?
L194: write -1 as superscript.
-The analysis of VOCs is one of the very interesting and important tests that were conducted in this research, but unfortunately, its results are placed in Supplementary Materials, which cannot be seen. I strongly recommend that the results of this analysis be shown in the results section. In addition, I could not see any of the results presented in Supplementary Materials, and all the tables should be added to the text of the manuscript.
-Color is one of the appearance characteristics of honey. Did the colors of the tested samples investigate?
L302-315: In which sample was the highest amount of VOCs observed? Compare the exact values of the samples.
Round 2
Reviewer 3 Report
The revised manuscript is suitable for publication in Foods Journal after making the following correction:
Tables S1 and S2: Change "Retetion time" to "Retention time".
Author Response
Dear Editor,
you can find in the file attached the reply to comments from the academic editor and reviewer 3.
Best Regards
Gabriele Capodaglio
